# The (Epistemological) *Power of Love*: From Pitirim A. Sorokin's *Integralism* to a 'Space for the Heart' in Scientific Methods

Licia Paglione

Department of Social and Political Sciences, Economics and Management Sophia University Institute, 50064 Figline e Incisa Valdarno, Italy; licia.paglione@sophiauniversity.org

**Abstract:** In the contemporary epistemological debate, it is possible to identify approaches whereby rational and sensory human faculties are not the sole essential dimensions in the knowing process. With its intuitional and empathic nature, love emerges as a specific way through which scientists may also know the world. In the 20th century, the Russian–American sociologist Pitirim A. Sorokin (1889–1968) was one of the scholars who highlighted the epistemological *power of love*. In his *integral epistemology,* the relevance of *Altruistic Creative Love* within the cognitive process is underlined. Love appears as an energy—linked to a specific intuitional human dimension called *supraconscious*—through which to know reality, which could integrate the empirical–sensory and rational–mindful dimensions. Following this line of thought, this article presents the thought of this scholar, outlining his general theory of knowledge and, in particular, highlighting the function of *Altruistic Creative Love* in the scientific method and analysing an original scientific dissemination activity *embedded* in the Sorokinian perspective that makes use of the arts, which can open new "eyes" and stimulate individual and social transformation.

**Keywords:** integral epistemology; Pitirim Sorokin; love; intuition; empathy; social transformation

## 1. Introduction

In the contemporary epistemological debate concerning the scientific approaches adopted to know reality, which is recognised as a complex object of analysis, we may encounter a line of thought whereby love emerges not only as an emotional state but also as a specific way that can allow us to analyse the world scientifically. In the 20th century, the Russian–American sociologist Pitirim A. Sorokin (1889–1968) was one of the scholars who drew attention to the epistemological 'power of love'" (Sorokin [1954] 2002). Through his proposal of an integral epistemology (Cipolla 2022; Jeffrey 2005), he reclaimed a 'space' for the energy of love in knowing processes and in the scientific methods connecting it with an intuitional cognitive "channel" (Abbottoni 2004, p. 70; Nichols 2005, p. 24; Cipolla 2022, p. 56), which is fundamental to integrate the empirical–sensory and rational–mindful dimensions (Abbottoni 2004, p. 70; Nichols 2005, p. 24) in order to acquire knowledge of the world. Following this line of thought, this article is structured into two parts. In the first part, Sorokin's general theory of knowledge and the connection between the search for truth and the creative and altruistic energy of love are presented (Section 1), highlighting the epistemological function of love in the knowing process (Section 2). In the second part, the article presents the analysis of a case of scientific dissemination activity *embedded* in the Sorokinian epistemological perspective, oriented to stimulate an original 'vision' of the world and move social transformations: an exhibition of paintings and works of art which express the relevance of love into an integral theory of knowledge and the activation of the intuitional cognitive "channel" in human beings (Section 3).

## 2. Pitirim A. Sorokin and the (Epistemological) Power of Love

The words 'power of love' are reminiscent of the audacious title of a famous book written by Pitirim A. Sorokin. In 1954, in the latter phase of his intellectual career and

shortly after the tragic Second World War, the Russian–American sociologist of an Orthodox culture, considered by many to be a classic albeit seriously unknown scholar (Marletti 2018, pp. 107–26; Mangone 2022), published a work entitled *The Ways and Power of Love* (Sorokin [1954] 2002). His intention was to focus on and reveal a precious element which might save human relationships and society, ensuring peace and generating a sense of fraternity. In this book, thanks to its altruistic and creative nature (Mangone 2020), love appears as a fundamental force for social integration and the flourishing of human beings. Moreover, in this work, Sorokin offered insights into the importance of love with respect to the acquisition of knowledge, in line with his original and more general theory of knowledge, where Sorokin suggests that moving beyond rationality and the physical senses, the super-rational and super-sensory dimensions may also engender a specific methodological and epistemological function which can facilitate the search for truth.

*2.1. Towards an Integral Epistemology*

The importance attributed to ultimate values and deep meanings induced Sorokin to develop an original theory of knowledge, which he referred to as "Integralism" (Sorokin 1941, 1956, 1957, 1958, 1966). He arrived at this idea as a result of a "blissful process of reintegration" (Sorokin 1963b), starting from a critical vision of mainstream "quantum-phrenic" and "numerolatrous" sociology in the mid-20th century, with its tendency to focus on numerical details and results and an obsession towards identifying analytical techniques and accumulating data (Sorokin 1956). The aim was to construct scientific methods through which he believed sociology would be able to comprehend reality in harmony with its ontological nature (Sorgi 1975, p. 19). In order to illustrate the nature of the Sorokinian perspective, it may be useful to summarise it, following the fundamental elements of a scientific paradigm (Ladyman 2007) and presenting it at the ontological, epistemological and methodological levels, highlighting its specific anthropological vision.

2.1.1. The Ontological Level: Reality as an Infinite and Dynamically Integrated X

Underlying Sorokinian *Integralism* is a conception whereby reality is perceived as an infinite totality, this being the result of a *coincidentia oppositorum*: a unity in which all contradictions are absorbed and from which, in turn, every difference originates (Abbottoni 2004, p. 68). More specifically, reality is described by Sorokin as an "infinite X of innumerable qualities and quantities: spiritual and material, personal and super-personal, temporal and timeless, spatial and spaceless, one and many, the very small as well as the small, the greatest as well as the greatest" (Sorokin 1958), corresponding to a living compossibility of elements located at three different levels.. These levels coincide with three different forms according to which reality is presented: the empirical–sensory form, the rational–mindful form and the super-sensory and super-rational form, i.e., made up of a composition of dimensions that lie beyond the reach of the physical senses and rationality. Integral reality comprises the totality of these three forms, held together by links that produce diverse degrees of integration.

2.1.2. The Epistemological and Methodological Level: The Integration of Three Dimensions

From Sorokin's point of view, knowledge of reality, thus perceived, needs an integral method (Sorgi 1985, pp. 117–18; Cipolla 2022) that is capable of surpassing a single dimension in opposition to all empiricist reductionism imposed by positivism, without denying its gnoseological value. The evolution of Sorokin's epistemological perspective is based not on the replacement of an empirical–quantitative (EQ) method with a logical–rational (LR) and, subsequently, an intuitive (I) method but in their integration, coinciding at the methodological level with a prospective expansion: from EQ to EQ + LR + I (Sorgi 1985, p. 135).

2.1.3. The Anthropological Level: Human Beings as Integral Beings

At the anthropological level, this expansion corresponds to the specific Sorokinian vision of human beings, defined by Sorokin as a "wonderful integral being" (Sorokin 1958, sect. 2),

i.e., a biological, rational and super-sensory and super-rational organism. In the integral human being, expansion results in the possibility of using the three diverse dimensions as cognitive "channels"—rationality, physical senses, and intuition—in an integrated manner.

Although all "channels" are of equal importance, Sorokin in his statement of "integral theory of truth and reality" (Sorokin 1941, pp. 746–64) accords primacy to intuition (Nichols 2005, p. 25; Abbottoni 2004, p. 70; Cipolla 2022, p. 35): it represents one of the privileged sources that permit access to certain truths, particularly at the ultra-sensitive level, offering a foundation for validity (Sorokin 1941, pp. 748–49). It is also the starting point for many scientific discoveries, assuming a decisive role in all creative activity and true knowledge. At the same time, Sorokin emphasises that intuition alone is insufficient, but it has to be assessed or verified by means of the other two previously mentioned "systems of truth". The three cognitive "channels", thus, offer information that forms a relationship of complementarity but also of mutual correction: a relationship of "epistemic correlation" (Sorokin 1956; Abbottoni 2004, p. 70; Nichols 2005, pp. 24–25). Thus, the integral system of truth ensures a higher degree of cognitive adequacy than the three systems of truth considered separately. The hypothesis proposed in this article is that in Sorokinian integral epistemology—linked to the supraconscious dimension of reality and an intuitional "channel" in human beings—another relevant element emerges: altruistic love. As Nichols (2005, p. 27) wrote, from the Sorokinian perspective "altruism takes knowledge provided by intuition". This aspect is particularly emphasised by the author in *The Ways and Power of Love* (Sorokin [1954] 2002). Here, love appears as closely connected to truth. However, in which sense? And what is Sorokinian love?

### 2.2. Sorokinian Love: Altruistic and Creative (Also Knowing) Energy

At the final phase of a long intellectual 'journey' undertaken by Sorokin (1963a), his interest in love emerges (Mangone and Dolgov 2019). It is seen by the author as a multidimensional phenomenon that acts as a creative energy integrating reality: "an inexhaustible universe in quality and quantity" that manifests itself in "many forms" (Sorokin [1954] 2002, p. 3)—seven—, some of which are visible, and some of which are not: religious, ethical, ontological, physical, biological, psychological, and social forms. As a sociologist, Sorokin focused on the last two forms of love: the psychological and the social. In the psychological form, love is seen as an altruistic experience that encompasses emotions, volition and intellect (Sorokin [1954] 2002, pp. 9–10). This form involves the identification of a lover with his/her beloved one without any loss of individuality, leading to a sense of freedom and the expansion of true individuality (Sorokin [1954] 2002, pp. 10–11). Sorokin suggests that, in this form, love is creative, engendering peace of mind and happiness (Sorokin [1954] 2002, p. 12). In the social form, love is viewed as corresponding to a meaningful interaction or relationship between two or more persons, where aspirations and goals are shared and supported (Sorokin [1954] 2002, p. 13). Sorokin considers this form of love as the "supreme and vital form of human relationship" (Sorokin [1954] 2002, p. 76), enabling the existence of a happy society and the realisation of the individual as an "integral being", as one may complete oneself in and through others (Sorokin [1954] 2002, p. 13). With regard to the empirical observation of love, the author proposes certain qualities as dimensions of an analytical model having "theoretical and practical significance" (Sorokin [1954] 2002, p. 19), allowing one to view the hidden part of an "iceberg" (Sorokin [1954] 2002, p. 3), which is not well known and has been underestimated by the contemporary positivist and "sensate culture" (Sorokin 1957). In particular, love may be observed taking into consideration five operational dimensions (Sorokin [1954] 2002, chap. II): intensity, extensivity, duration, purity, and adequacy. The first dimension—intensity—reveals the greatness and preciousness of what is freely given. The second dimension—extensivity—ranges from an exclusive love for oneself to love for humanity, with all living creatures and the entire universe. The third dimension—duration—may vary from a brief moment, as may occur in a heroic act, to years or the entire course of a lifetime, as in a mother's experience of caring for her child. The fourth dimension—purity—relates to the logic and the motivations that

animate people and "ranges from the love motivated by love alone—without the taint of a "soiling motive" of utility, pleasure, advantage or profit—down to the "soiled love", as a means to realise utilitarian and hedonistic goals (Sorokin [1954] 2002, p. 17). The last dimension—adequacy—concerns the more or less congruent relationship between the subjective motivations of an act of love and its objective consequences. In accordance with Russian Orthodox spirituality and philosophy, which profoundly influenced Sorokin's thought (Abbottoni 2004; Ponomareva 2011; Tiryakian 1988; Nichols 2012), Altruistic Love is closely connected to two other great values: beauty and truth. However, love and truth are "inseparable" (Sorokin [1954] 2002, p. 6), and this influences the Sorokinian epistemological perspective, introducing a 'space' for love as a fundamental *powerful energy* and also in the process of the growth of scientific knowledge. How does this occur?

### 3. The *Power of Love* in the Search for Truth (and *Vice Versa*): Mutual Support and Transformation

Sorokin notes that "Love is viewed as the essence of goodness inseparable from truth and beauty. All three are unified aspects of the Absolute Value" (Sorokin [1954] 2002, p. 6), and all three have the same integrative function, uniting what is separate (Sorokin [1954] 2002, p. 6). Analysing the Sorokinian perspective, it is clear that the connection between love and truth is bidirectional. On the one hand, love contributes towards a search for truth: "the power of love greatly reinforces the power of truth and knowledge" (Sorokin [1954] 2002, p. 78) because love is "one of the surest and most efficient methods of cognition and the most fruitful way to truth and knowledge" (Sorokin [1954] 2002, p. 78) and "[...] contains within itself cognitive elements that enrich us with intuitive or rational, or even empirical cognition or truth" (Sorokin [1954] 2002, p. 31). On the other hand, truth supports the efficacy of Love: "[...] an increase in the number of these heroes of truth [...] leads indirectly to an increase in the production of love" (Sorokin [1954] 2002, p. 41). More specifically, Sorokin highlights that love and truth are interchangeable; the former may be transformed into the latter, and *vice versa*. Love and truth are, in fact, two diverse and yet mutually influenceable elements, this being a factor which Sorokin refers to as "support" (Sorokin [1954] 2002, p. 318), and they may collaborate in attempts to realise a particular purpose. To emphasise this further, reflecting on a sort of 'interchangeability' occurring between subjects who are active within these human spheres and between those dedicated to goodness (Love) and those dedicated to the pursuit of truth, Sorokin explains that "Their external garb changes, but their real function remains the same: they act as power stations generating the energy of love for humanity" (Sorokin [1954] 2002, p. 40). The idea of a mutually supportive connection between love and truth seems to suggest that, although these values may also exist independently, their power to unify, integrate, and harmonise depends on the degree to which they are mutually integrated: searching for truth could be more creative when it is nourished by more love. This position is further clarified by the Sorokinian idea that "[...] the possibility of the transformation of one of these 'energies' into the other two follows logically from the hypothesis [...]" (Sorokin [1954] 2002, p. 30). The possibility of transforming one of these energies into the others means that "all those who enriched humanity with truth and beauty have also contributed to a more efficient production of love" (Sorokin [1954] 2002, p. 39). It, thus, means that a scientist, while creating truth, can contribute, at the same time, to creating love. And *vice versa*: "A 'loving heart' often knows or feels instinctively (that is, intuitionally) when and what is wrong and what is right for the loved one" (Sorokin [1954] 2002, p. 29). Moreover, Sorokin explains that the transformation of truth into love and *vice versa*, is not total, but it "always gives an efficiency below 100 per cent" (Sorokin [1954] 2002, p. 31) due to the influence of the "qualitative-quantitative magnitude of each energy" (Sorokin [1954] 2002, p. 31), i.e., according to the different combinations of the five dimensions proposed for the analysis of love:

"[. . .] *the intenser, the purer, more extensive, durable, and adequate the given energy of truth or beauty is, the greater tends to be the percentage of its efficient transformation into goodness (love) energy; and vice versa*". (Sorokin [1954] 2002, pp. 31–32)

From Sorokin's perspective, love (or goodness) emerges as a fundamental part of the integral system of truth, expressing creative power in cognitive terms. The gnoseological efficacy of this method—in line with the Russian philosophical tradition, in which Sorokin is fully immersed—is linked to the degree of involvement in the cognitive process of a human being in its entirety and is proportional to the fullness with which this integrity is realised. Following this idea, the author seems to draw attention to where this happens: the interiority of the knowing subject. It is considered a 'centre' in which reason, will, feeling, and the entire complex of a spirit converge into a single living unity (Kireevskij 1978, pp. 242–43). Thus, between a 'knowing' subject (an integral and internally fully integrated being) and the object of knowledge ('integral reality'), there is a sort of continuity engendered by sharing the same ontological structure and common belonging to a totality that contains both. From the Sorokinian epistemological perspective, a specific 'space' for love appears in the knowing process, connecting the subject and the object, a human being and reality, and interiority and the world (Table 1).

**Table 1.** Sorokinian *integralism* and the epistemological and methodological level as a 'space' for love.

| Sorokinian Integralism | | |
| --- | --- | --- |
| **Ontological Level** | **Epistemological and Methodological Level**<br>←   **Space for Love**   → | **Anthropological Level** |
| Reality as an infinite and dynamically integrated X. Three forms: empirical–sensory, rational and super-rational, and super-sensory | Integral method EQ + LR + I (Sorgi 1985, p. 135) | Human being as an integral being with three cognitive "channels": empirical–sensory, rational–mindful, and intuitional |

## 4. From *Sorokinian Integralism* to the *Into. Knowing from the Heart* Exhibition

Following the Sorokinian perspective of knowledge, sociological imagination may activate itself to stimulate both rational and sensorial dimensions and the intuitive one, involving all human cognitive dimensions and connecting them with external reality, i.e., the world. In this process, love—as part of an intuitional form of knowledge—could also express itself and integrate all cognitive "channels" of knowledge. To describe this perspective, artistic creativity could also offer a specific contribution. This article presents an analysis of a scientific dissemination activity *embedded* in the Sorokinian theory of knowledge: an artistic-epistemological project, entitled *Into. Knowing from the Heart.* It was an exhibition of paintings inspired by Sorokinian integralism, which was recently developed to stimulate reflection about forms of integral knowledge. During a recent international academic conference on the subject of love, critically entitled *The Movement of Agape*[1], promoted in Italy (Florence) by the Sophia University Institute, apart from the relative lectures and academic discussions, the participants were invited to become more deeply and critically acquainted with the role of the intuitional dimension and love in the knowing process through a presentation of works of art and writings. Ten oil paintings presented in the hall of the Institute (Figure 1), accompanied by epistemological texts presented by various scholars—Sorokin included—induced visitors to reflect at both the rational and also at sensory and intuitional level. Through the paintings, in fact, the visitors were stimulated in three ways: in rational thought, in physical senses, and also in their intuitional dimension to distinguish spiral forms, created in different colours and with a variety of materials, depicting centripetal and centrifugal movements as an abstract

representation of integral knowledge, linking the human interior sphere and reality. In her critical presentation, written for the exihibition, Serena Meattini, a researcher and expert in the philosophy of art, noted that the following:

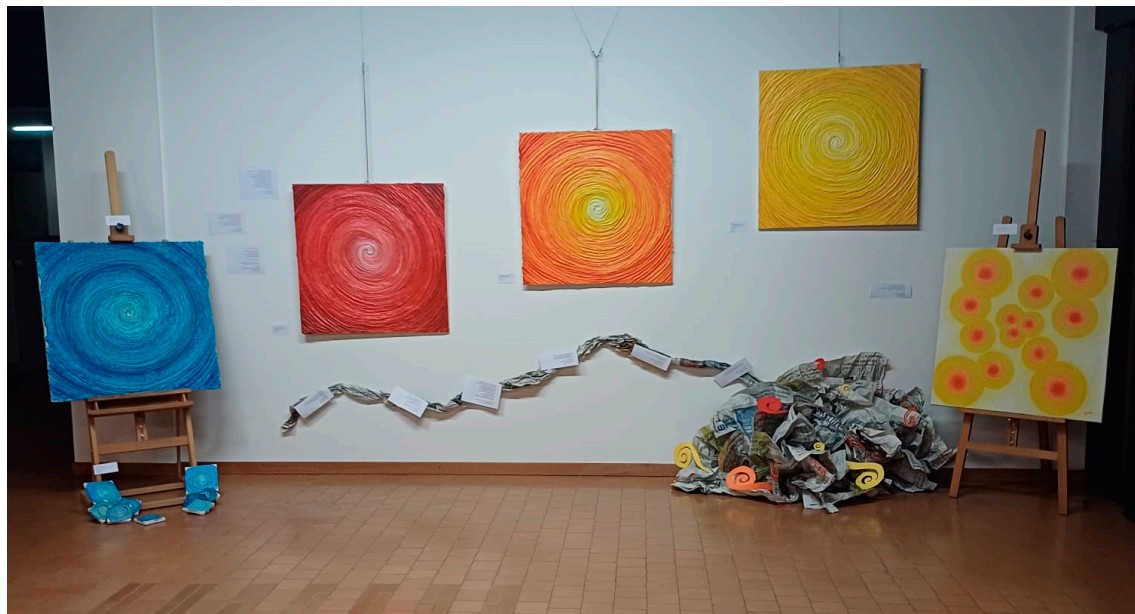

**Figure 1.** Artistic works presented at the *Into. Knowing from the Heart* exhibition (partial view).

> "The dynamism of the spiral motif is characterised by a dual, centrifugal and centripetal movement reflecting an expansion of the self into the world, but also by the inclusion of the external world—and, in particular, the dimension of other beings—within the self".

This dynamism may be associated with the Sorokinian integral approach that "perceives the change of any sociocultural phenomena as being the result of the combination of external and internal forces" (Mangone 2018). The spiral forms which appear in the paintings, and which might be discerned as elements reflecting an almost 'obsessive' vision, highlight the evolution of the artist's style and her dedication to offering a profound illustration of the 'knowing' dynamic and asymmetric movements of the hand, which may be recognised in the spirals, following a rhythm of unpredictable human complexity (or Sorokinian integrality?). As Meattini wrote in her comment:

> "The asymmetric forms fully correspond to the rhythm of the hand that freely creates. The risk of an unpredictable human complexity remains present and is reflected in the free construction of forms which may stimulate visions of an implicit dynamism."

The purpose of the epistemological texts situated close to the paintings, written by scholars and experts from various disciplines (sociology, philosophy and natural sciences in particular), was to stimulate rational reflection on the part of visitors. The texts presented, composed by the Italian–German philosopher Romano Guardini (1885–1968), the German sociologist Georg Simmel (1858–1918) and the Romanian physicist Basarab Nicolescu (1942-) and others, were selected by colleagues of the artist and researchers in various scientific fields. Highlighting the consonance of thought between these authors and Sorokin had the aim of establishing a focus on the shared need in epistemology to define forms of integral knowledge in which super-rational and super-sensory dimensions—particularly love—are essential elements. Some images of the paintings may be explained better. For example, in Figure 2, the trilogy entitled *Integrality* (2021–2022) is aimed at highlighting the three Sorokinian cognitive "channels" in human beings. Each painting represents one of these

forms (rational–mindful, empirical–sensory, and intuitional), which is indicated by the titles: 'mind', 'hand', and 'heart'.

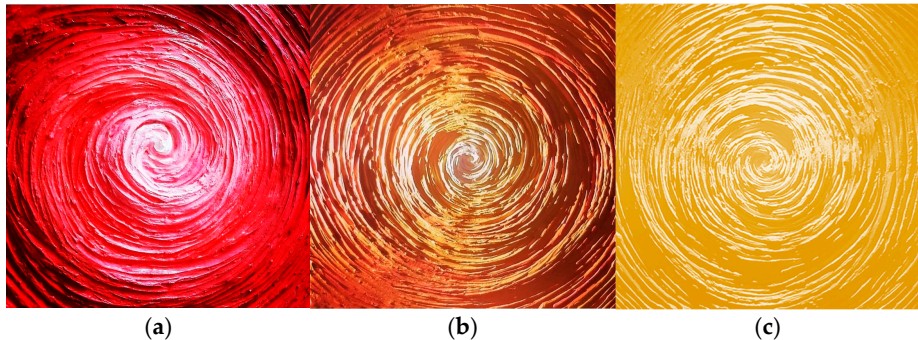

|     |     |     |
| :-: | :-: | :-: |
| (**a**) | (**b**) | (**c**) |

**Figure 2.** *Integrality* (2021–2022). (**a**) *Knowing Heart*, (**b**) *Knowing Mind* (**c**), and *Knowing Hand*.

An incomplete painting without any colour, presented (in Figure 3) at the centre of the exhibition space, invites observers to think and reflect on how they may be internally stimulated. The title of this painting is Forever (2024), reminding us of the Sorokinian idea of the importance of the infinite duration of love, which may be well-rooted in the past and may engender future transformation at both the individual and social level.

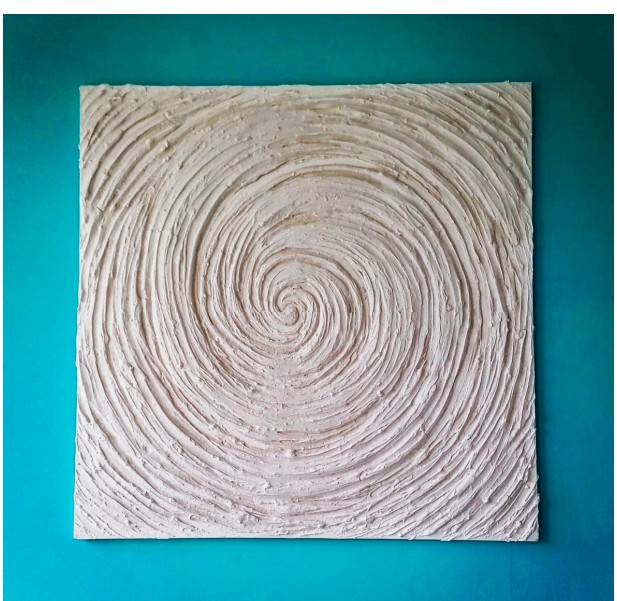

**Figure 3.** *Forever* (2024).

The final work of art was not a painting but an 'installation' created using old newspaper sheets and spiral forms produced with coloured paper (Figure 4). The title, *Transformation* (*between Interiority and World*) (2024) reminds us of the stimulating results of the presence of love in individuals and in society and also the interaction and influence possibly occurring between a personal and collective reality, once again highlighting the Sorokinian link between human interiority and the external socio-cultural force (Mangone 2018).

The relevance of the super-rational and super-sensory dimensions emerge through the works presented at the exhibition. As Serena Meattini writes in her critical presentation,

"Through theoretical reflection and visual stimulation, between a recounting of elements and a chance to view them, the *Into.* exhibition invites us to dwell specifically on an intermediate space that reminds us of the genetic locus of truth in asymmetrical and unpredictable beauty which can be fully grasped only by affective and intuitional knowledge."

In this sense, the *Into.* exhibition seems to represent an experimental space for love in the knowing process, where rationality, the physical senses, and the intuitional dimensions are integrated and—in Sorokinian terms—as a whole entity, the visitor may acquire an integral vision of the world.

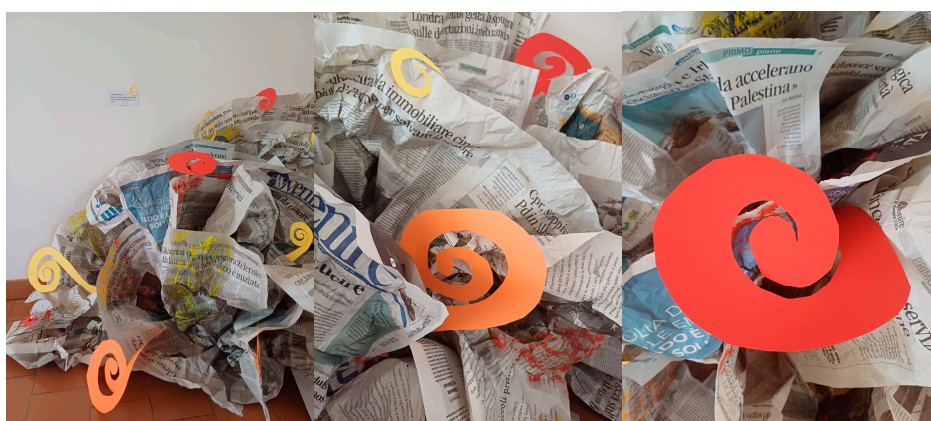

**Figure 4.** *Transformation* (between interiority and world) (2024).

### 5. Conclusions. *Knowing from the Heart*: From Sorokin, a Way to Integrally Know and Move Individual and Social Transformation

Sorokin's perspective and, in particular, his integral epistemology (Cipolla 2022) highlight a specific function of intuitional dimensions and, within this, of *Altruistic Love* as a fundamental force in the process of acquiring scientific knowledge. From his perspective, Sorokin opens up a 'space' for love as a creative energy that integrates all human dimensions. The creative energy of love seems to connect human beings who seek the truth, such as scientists and researchers ('knowing' subjects), and reality—or the world (the 'known' object)—at many levels, in line with contemporary inter- and transdisciplinary scientific approaches (Galluzzi and Paglione 2024). Through its global and wide-ranging vision, Sorokin's integralism allows us to imagine original ways of generating and disseminating scientific knowledge, "opening new eyes"—as Romano Guardini would say[2]. The Sorokinian theory allows us to give a 'space' for love in the knowing process and in scientific methods—showing the potential cognitive *power of love*—to 'see' the complexity of reality integrally and at every level, which, as highlighted by Basarab Nicolescu (1996)[3], may stimulate individual and social transformation (Cataldi and Iorio 2023), following an approach that conceives changes occurring in any sociocultural phenomena as the result of the combination of external and internal forces (Mangone 2018).

**Funding:** No external funding was granted for this research.

**Institutional Review Board Statement:** Not applicable.

**Informed Consent Statement:** Not applicable.

**Data Availability Statement:** The original contributions presented in the study are included in the article. further inquiries can be directed to the corresponding author.

**Conflicts of Interest:** The author declares no conflict of interest.

## Notes

[1]    The title of the conference mentions love as agape, a concept that is not entirely consistent with the Sorokinian concept of love. In the conference, however, the concept of agape was introduced and critically discussed in relation to other forms of love.

[2]    The philosopher Romano Guardini (1950) connects the heart with the eyes and in this way introduces the relevance of the human interior dimension in our endeavour to comprehend the world. His comment, presented at the *Into.* exhibition, was "[…] the roots of the eye are in the heart" (selected by Andrea Galluzzi, translated by the author).

[3]    In his *Manifesto of Transdisciplinarity* (1996) the physicist Basarab Nicolescu stated: "[…] when our perspective of the world changes, the world changes" (selected by Andrea Galluzzi).

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
