# Peer review of "The (Epistemological) Power of Love: From Pitirim A. Sorokin’s Integralism to a ‘Space for the Heart’ in Scientific Methods"

_socsci, doi:10.3390/socsci13090482_

Round 1

Reviewer 1 Report

Comments and Suggestions for Authors

Usually studies on Sorokin always start with the same clssic topic. Sorokin was an original sociologist because he broke the dominant theoretical model of the mid-twentieth century. In fact, at that epoch, social sciences were influenced by conflict theories, studies on ideology, and examinations of rationalization processes: something of different in relation to the proposal of Sorokin (altruistic love). 

The essay you submitted to me starts from an original point of view. Sorokin's sociology of love is not a political rupture, but an epistemological one. His theory is not a simple paradigm shift, but an attempt to go deeper into human nature and social relationships. For this reason, I have appreciated that among the sources of the paper there are the studies of T. Sorgi, because I think he was the best scholar of Sorokin (in many other studies Sorgi is forgotten).

I think is good the bridge that the paper does between sociology and visual art, in relation to the activation of the "intuitional cognitive channel".  This connection is not trivial in a historical phase marked by multidisciplinarity. Not only that: I believe that this approach can encourage further development in terms of the application of social theories to artificial intelligence.

Author Response

I accept your comments.

The article now includes some modification about references, incorrect words and other aspects to specify better the central idea and connect better the first and the second part.

Reviewer 2 Report

Comments and Suggestions for Authors

The article has an ambitious goal, which is to use Sorokin’s theory of sociology of knowledge, trying to underline the functions of altruistic creative love as a “channel” to read social reality. This goal is not achieved because the article is based on some elements that are critical in the use of Sorokin’s theories and that necessarily need to be corrected. The most important and significant ones are highlighted below:

1) it is not possible to have Sorokin's theory of sociology of knowledge as a presupposition by limiting oneself to only one text (Sorokin, P.A. 1958. “Integralism is My Philosophy.” In This is my philosophy. Twenty of the world’s outstanding thinkers reveal the deepest meaning they have found in life, edited by Whit Burnett, 180-189. London: George Allen & Unwin), leaving aside all the vast scientific production in which this theory is developed and clarified    (a few examples: Sorokin, P.A. 1962. Society, Culture, and Personality: Their Structure and Dynamics. A System of General Sociology. New York: Cooper Square Publishers - Original edition of Harper & Brothers, 1947; Sorokin, P.A. 1963. Sociology of my Mental Life. In P.J. Allen (ed), Pitirim A. Sorokin in Review (3-36). Durham: Duke University Press);

2) Sorokin proposes a system of knowledge that integrates the three forms of knowledge (empirical-sensory, rational-mindful, and supersensory-superrational) linked together by a causal-significant link (they are not complementary as stated in the article);

3) Sorokin himself defines his system of knowledge as a “sociological integralism” method (Sorokin, Pitirim A. 1989. “On Sorokin.” Science in Context 3: 299-302; Ford, J.B. 2017. “Sorokin’s Methodology: Integralism as the Key.” In Sorokin and civilization: a centennial assessment, edited by Joseph B. Ford, Michel P. Richard, and Palmer C. Talbutt, 83-92. New York: Routledge - Original edition of Transaction Publishers, 1996) and, therefore, cannot be considered an epistemology;

4) Regarding Sorokin’s concept of love as discussed in the article, two errors are highlighted: a) the title of the book is reported incorrectly, this is the correct title: The Ways and Power of Love); b) attributing the adjectives altruistic and creative are attributed incorrectly, in fact, in this book Sorokin despite referring to the ways and power of love, does not provide a definition of the word «love» (Eckhardt W., 1983, «Pioneers of peace research III: Pitirim A. Sorokin: Apostle of love», International Interactions: Empirical and Theoretical Research in International Relations, 10-2, p. 147-177; Mangone, E. 2020. Pitirim A. Sorokin’s Contribution to the Theory and Practice of Altruism, “Revue européenne des sciences sociales/European Journal of Social Sciences”, 58(1), pp. 149-175) making the words «love» and «altruism» interchangeable. It is an energy that creates/generates (hence the adjective creative) all the actions that produce and maintain the psychological and/or physical good of oneself and others through altruism;

5) There are terms within the article that are not attributable to Sorokin as for example: “channels”, Sorokin speaks of vehicles and forms; sensorial, rational and supra-intuitional, are not the terms that Sorokin uses for his system of knowledge of reality (these are the terms used: empirical-sensory, rational-mindful, and supersensory-superrational) or sensualist culture (correct terminology: sensate culture).

In light of these considerations, therefore, the content of the last paragraph of the article (3. From Sorokinian Integralism to the Into. Knowing from the Heart exhibition: a presentation of artistic works that depict the connection between the human interior dimension and the world) is not understood, which tries to connect Sorokin's concept of love with the comments on the works present in an exhibition, since Sorokin's concept of love is not a form of knowledge but a creative force (which should move the world towards transformations) and is not agapic love (there is a reference to an international conference entitled, The Movement Agape). In order to think about publishing the article, all the critical issues highlighted must be resolved and above all the connection between the first part of the article and the last paragraph must be specifically clarified.

Comments on the Quality of English Language

The English is readable well, but there are terms that need to be corrected because they were not used by Sorokin in his works.

Author Response

Comments and replies:

1) it is not possible to have Sorokin's theory of sociology of knowledge as a presupposition by limiting oneself to only one text (Sorokin, P.A. 1958. “Integralism is My Philosophy.” In This is my philosophy. Twenty of the world’s outstanding thinkers reveal the deepest meaning they have found in life, edited by Whit Burnett, 180-189. London: George Allen & Unwin), leaving aside all the vast scientific production in which this theory is developed and clarified    (a few examples: Sorokin, P.A. 1962. Society, Culture, and Personality: Their Structure and Dynamics. A System of General Sociology. New York: Cooper Square Publishers - Original edition of Harper & Brothers, 1947; Sorokin, P.A. 1963. Sociology of my Mental Life. In P.J. Allen (ed), Pitirim A. Sorokin in Review (3-36). Durham: Duke University Press);

--> As suggested by the reviewer, other important bibliographical references of the author have been more explicitly included in the article (text and reference list). In particular:

(Sorokin 1966) Sorokin, Pitirim A. 1966. Sociological Theory of Today. San Francisco: Harper and Row.

(Sorokin 1963) Sorokin, Pitirim A. 1963. Sociology of My Mental Life. In Pitirim A. Sorokin in Review: The American Sociological Forum. Edited by Allen Philip J. Duke University Press: Durham, NC. pp. 4-36.

(Sorokin 1941) Sorokin, Pitirim A. 1941. Social and Cultural Dynamics. Vol. IV. New York: Dutton.

Other references concerning the theory of knowledge of the author was cited in the article:

(Sorokin 1957) Sorokin, Pitirim A. 1957. Social and Cultural Dynamics. Boston: Porter Sargent.

(Sorokin 1956) Sorokin, Pitirim A. 1956. Fads and Foibes in Modern Sociology and Related Sciences. Chicago: H. Regnery Co.

Some critical articles was introduced. In particular:

(Nichols 2005) Nichols, Lawrence T. 2005. Integralism and positive psycology: a comparison of Sorokin and Seligman. The Catholic Social Science Review 10: 21-40.

(Jeffrey 2005) Jeffrey, Vincent. 2005. Pitirim A. Sorokin's Integralism and Public Sociology. The American Sociologist XXXVI (3-4): 66-87.

2) Sorokin proposes a system of knowledge that integrates the three forms of knowledge (empirical-sensory, rational-mindful, and supersensory-superrational) linked together by a causal-significant link (they are not complementary as stated in the article);

--> In the article, line 107-124, this link is explained through the concept of ‘epistemic correlation’, following a terminology of the author (Fads and Foibes, 1956) and his critical scholars, quoted in the text: Nichols 2005, pp. 24-25, Abbottoni 2004, p. 70.

3) Sorokin himself defines his system of knowledge as a “sociological integralism” method (Sorokin, Pitirim A. 1989. “On Sorokin.” Science in Context 3: 299-302; Ford, J.B. 2017. “Sorokin’s Methodology: Integralism as the Key.” In Sorokin and civilization: a centennial assessment, edited by Joseph B. Ford, Michel P. Richard, and Palmer C. Talbutt, 83-92. New York: Routledge - Original edition of Transaction Publishers, 1996) and, therefore, cannot be considered an epistemology;

--> Following the reviewer's suggestions, the article makes increased use of terms ‘theory of knowledge’ and ‘method’.The article uses also the term ‘integral epistemology’, following Cipolla 2022, Jeffrey 2005 for instance. The missing references are now in the text.

4) Regarding Sorokin’s concept of love as discussed in the article, two errors are highlighted: a) the title of the book is reported incorrectly, this is the correct title: The Ways and Power of Love); b) attributing the adjectives altruistic and creative are attributed incorrectly, in fact, in this book Sorokin despite referring to the ways and power of love, does not provide a definition of the word «love» (Eckhardt W., 1983, «Pioneers of peace research III: Pitirim A. Sorokin: Apostle of love», International Interactions: Empirical and Theoretical Research in International Relations, 10-2, p. 147-177; Mangone, E. 2020. Pitirim A. Sorokin’s Contribution to the Theory and Practice of Altruism, “Revue européenne des sciences sociales/European Journal of Social Sciences”, 58(1), pp. 149-175) making the words «love» and «altruism» interchangeable. It is an energy that creates/generates (hence the adjective creative) all the actions that produce and maintain the psychological and/or physical good of oneself and others through altruism;

--> The title of the book was corrected.

The concept of love is presented as energy or force in various points (in red in the text) and described following the Sorokinian multidimensional idea (seven forms) and integrating function.

5) There are terms within the article that are not attributable to Sorokin as for example: “channels”, Sorokin speaks of vehicles and forms; sensorial, rational and supra-intuitional, are not the terms that Sorokin uses for his system of knowledge of reality (these are the terms used: empirical-sensory, rational-mindful, and supersensory-superrational) or sensualist culture (correct terminology: sensate culture).

--> The term ‘channel’ comes from some critical scholars: Nichols 2005, Abbottoni 2004, Cipolla 2022. The references (previously missing) are now in the text. Incorrect terms highlighted by the reviewer have been corrected.

In light of these considerations, therefore, the content of the last paragraph of the article 3. From Sorokinian Integralism to the Into. Knowing from the Heart exhibition: a presentation of artistic works that depict the connection between the human interior dimension and the world) is not understood, which tries to connect Sorokin's concept of love with the comments on the works present in an exhibition, since Sorokin's concept of love is not a form of knowledge but a creative force (which should move the world towards transformations) and is not agapic love (there is a reference to an international conference entitled, The Movement Agape).

In order to think about publishing the article, all the critical issues highlighted must be resolved and above all the connection between the first part of the article and the last paragraph must be specifically clarified.

--> I tried to modify and connect the parts better, analysing the exhibition as a case of a scientific dissemination activity embedded in Sorokinian thought and emphasising the concept of love as a creative force.

In addition, a note was inserted to explain that in the conference the concept of agape was introduced in a critical way, not only as agape.

Round 2

Reviewer 2 Report

Comments and Suggestions for Authors

The revision added from author/authors are sufficient, but I permet to suggest to author/authors in the future - if the interest on Sorokin will continue - to consider the svholars recognized internationally as experts on Sorokin's thoughtand non-scholars who spradically (and within the national borders) and with much "imagibation" have written about this author.